# Novel Co-Polyamides Containing Pendant Phenyl/Pyridinyl Groups with Potential Application in Water Desalination Processes

**DOI:** 10.3390/polym17020208

**Published:** 2025-01-15

**Authors:** Carolina Arriaza-Echanes, Claudio A. Terraza, Alain Tundidor-Camba, Loreto Sanhueza Ch., Pablo A. Ortiz

**Affiliations:** 1Centro de Nanotecnología Aplicada, Facultad de Ciencias, Ingeniería y Tecnología, Universidad Mayor, Camino La Pirámide 5750, Huechuraba 8580745, Chile; carolina.arriaza@mayor.cl; 2Research Laboratory for Organic Polymers (RLOP), Department of Organic Chemistry, Pontificia Universidad Católica de Chile, Santiago 7820436, Chile; cterraza@uc.cl; 3UC Energy Research Center, Pontificia Universidad Católica de Chile, Santiago 7820436, Chile; 4Department of Chemical & Biological Engineering, The University of Alabama, Tuscaloosa, AL 35487-0203, USA; atundido@uc.cl; 5Núcleo de Química y Bioquímica, Facultad de Ciencias, Ingeniería y Tecnología, Universidad Mayor, Camino La Pirámide 5750, Huechuraba 8580745, Chile; loreto.sanhueza@umayor.cl; 6Escuela de Ingeniería en Medio Ambiente y Sustentabilidad, Facultad de Ciencias, Ingeniería y Tecnología, Universidad Mayor, Camino La Pirámide 5750, Huechuraba 8580745, Chile

**Keywords:** co-polyamides, membranes, water desalination, pyridinyl pendant group

## Abstract

This study explores the development and evaluation of a novel series of aromatic co-polyamides featuring diverse pendant groups, including phenyl and pyridinyl derivatives, designed for water desalination membrane applications. These co-polyamides, synthesized with a combination of hexafluoroisopropyl, oxyether, phenyl, and amide groups, exhibited excellent solubility in polar aprotic solvents, thermal stability exceeding 350 °C, and the ability to form robust, flexible films. Membranes prepared via phase inversion demonstrated variable water permeability and NaCl rejection rates, significantly influenced by the pendant group chemistry. Notably, pyridinyl-substituted membranes achieved water fluxes up to 17.7 L m^−2^ h^−1^ and a NaCl rejection of 37.3%, while phenyl-substituted variants provided insights into the interplay of hydrophobicity and porosity. These findings highlight the critical role of pendant group functionality in tailoring membrane performance, offering a foundation for further structural modifications to enhance efficiency in water treatment technologies.

## 1. Introduction

In the coming decades, global freshwater consumption is expected to increase significantly, driven by rapid population growth, socio-economic development, urbanization, and the increasing need for water in the agricultural, industrial and energy sectors [1]. In addition, climate change is likely to lead to a significant reduction in water resources in several regions of the world [2].

Currently, a high percentage of the world’s population lives in water-scarce regions, and it is estimated that by 2030, this number will reach 2.7 billion people [3]. Although the Earth’s surface water amounts to approximately 1386 million cubic kilometers, only 1% of this resource is suitable for human use. Unfortunately, most of the available water has a high salt content (oceans and ice) [4]. Therefore, desalination and wastewater recycling have become necessities in recent years [5]. Among the various processes for obtaining water suitable for human use and consumption, membrane-based separation methods are the most promising because of their simplicity, cost-effectiveness, and environmental sustainability. However, when this technology treats extremely concentrated water, it significantly increases energy consumption [6]. Currently, membrane-based water desalination technology focuses on minimizing the rejected concentrate, controlling and preventing membrane fouling and, above all, improving membrane performance by balancing salt rejection and permeate flux [7]. However, it must always be kept in mind that an increase in permeate water flow rate results in a decrease in salt rejection, and therefore, in separation efficiency.

Some of the methods to improve the performance of membranes include functionalizing their surface by doping with nanoparticles (NPs), changing the surface morphology, and grafting functional groups [8,9]. Nonetheless, the most successful strategy to date has been the chemical and physical modification of membranes by incorporating functional groups into membrane-forming polymers, improving water flux, salt rejection, fouling resistance [9], and chlorine tolerance [10,11].

Among the parameters that can predict the performance of a membrane, porosity occupies a special place, since, regardless of variations in the radius or in the random network of membrane pores, porosity is always an important factor that determines its performance [12,13]. This is evident when considering that the pore size and the tortuosity generated by the pore network can either increase or reduce the passage of water and salts through the membrane.

Porosity is not the only factor that determines the performance of a membrane; its hydrophilicity is also important. Although experimental results show that hydrophilic pores favor higher permeability [14,15,16], completely hydrophilic membranes have poor stability in an aqueous environment and, therefore, cannot fulfill their purpose in the long term [17]. The membranes most commonly used today in water desalination have a certain degree of hydrophobicity. This is the case of thin-film composite (TFC) membranes, where the surface layer is made of an aromatic polyamide. In this type of membrane, the transport of water molecules occurs mainly by molecular diffusion. Since water is a small molecule and hydrophilic sites (amide groups) are abundant in the polyamide [18], membrane hydrophilicity has become a common challenge in water treatment applications.

Since the introduction of polyamide (PA) TFC in 1981 [19], this type of membrane has been the most widely used in water desalination processes. However, although these materials have demonstrated high water flux and salt rejection, their hydrophobic aromatic groups and high degree of cross-linking inherently limit the water flux, which in turn increases the cost of desalination [20]. Therefore, the search for alternative membrane preparation methods may be key to achieving better performance in water desalination processes [21].

A common method for preparing porous membranes for water treatment, particularly for ultrafiltration (UF), is the phase inversion method. This method consists of dissolving the polymer in a solvent, then pouring the homogeneous solution onto a suitable substrate and, finally, immersing the substrate with the polymer in a coagulation bath (non-solvent) for a specific duration. As a result of the exchange between solvent and non-solvent, precipitation occurs and a membrane film is formed. The excess solvent is subsequently removed by thoroughly washing the membrane with water [22]. Among the many conditions that must be considered to obtain phase inversion membranes, the solubility of the polymer and target solvent are key parameters for accurately modulating the microstructure of the polymeric membrane [23]. Therefore, achieving high polymer solubility and selecting the appropriate solvent are crucial factors to consider when preparing these types of membranes [24].

When designing new polymeric materials, one of the relevant factors to consider is solubility. This property is directly linked to the various stages that lead to the material’s final application. Notably, solubility plays an important role during synthesis and purification. It is expected that during the growth of the polymer chains, they remain in solution, which promotes an increase in molecular weight [25]. Solubility is also crucial during the characterization stage, as most techniques require the sample to be soluble in order to be performed [26]. Furthermore, in the processing stages, high solubility helps minimize manufacturing costs, as many processes require high temperatures and mechanical stress to achieve the final shape of the material for its intended application [27].

Among the strategies to achieve good solubility in polymeric materials is the inclusion of flexible groups such as oxyether, thioether, sulfone, or sp^3^-carbon atoms [28,29,30]. These groups have been successful in increasing the solubility of high-performance polymers such as aromatic polyamides. Another modification that can be incorporated is the presence of 1,3-disubstituted aromatic groups [29,31]. These groups disrupt the symmetry of the polymer chains, reducing their packing. Additionally, the introduction of bulky pendant groups reduces packing, substantially improving solubility [32,33,34].

Based on these principles, our research group has designed, synthesized and characterized three new soluble aromatic co-polyamides. These polymers were used to fabricate porous membranes via the phase inversion method. Finally, these membranes were characterized and preliminarily evaluated based on their NaCl rejection and water flux to assess their potential for water desalination applications.

## 2. Materials and Experimental Methodology

### 2.1. Materials

3,5-Dinitrobenzoyl chloride (DNC), 4,4-oxydianiline (ODA), hydrazine hydrate (H_2_NNH_2_, 80%*w*/*w*), palladium/charcoal activated (Pd/C, 10%*w*/*w* Pd), ethanol, methanol, tetrahydrofuran (THF), diethyl ether (Et_2_O), chloroform, *n*-hexane, *N*,*N*-dimethylformamide (DMF), *N*,*N*-dimethylacetamide anhydrous (DMAc), dimethylsulfoxide (DMSO), acetone, and deuterated dimethylsulfoxide (DMSO-*d_6_*) were obtained from Sigma-Aldrich (Milwaukee, WI, USA). 4,4′-(Perfluoropropane-2,2-diyl)dibenzoyl chloride was synthesized from 5 g of 4,4′-(perfluoropropane-2,2-diyl)dibenzoic acid by treatment with thionyl chloride at 80 °C for 3 h [35]. The reaction product was precipitated in *n*-hexane and purified by recrystallization in the same solvent (92% yield).

### 2.2. Monomer Synthesis

The diamine monomers were obtained following the two-step methodology previously reported [36]. In the first stage, the halide belonging to the 3,5-dinitrobenzoyl chloride is exchanged for the respective amine, obtaining the three desired dinitro-amide intermediates. The experimental setup (Appendix A) and the results of the spectroscopic characterization of the dinitro intermediates (Appendix A) are provided in the Appendix A. In the second stage, the intermediates were reduced using Pd/C and hydrazine, allowing for the production of the three aromatic diamines. The diamines synthesis route is described in Figure 1, and the experimental setup of the second synthesis stage is shown in Appendix A of the Appendix A. The results of the spectroscopic characterization of the diamines (Appendix A) are provided in the Appendix A.

### 2.3. Polymer Synthesis

The polymer synthesis was carried out in a manner analogous to that previously described [36]. In a 50 mL three-neck round-bottom flask, equipped with a mechanical stirrer and under constant N_2_ flow (Appendix A), a mixture of ODA (1.0012 g, 5 mmol) and PyMDA (1.2115 g, 5 mmol) was completely solubilized in 10 mL of anhydrous DMAc. Then, a mixture composed of LiCl (1.000 g, 23.6 mmol, 5%*w*/*v*) and 5.0 mL of anhydrous DMAc was added to the previous solution under stirring until the complete solubilization of the salt. The resulting mixture was cooled down to −10 °C using an acetone-ice bath, followed by the addition of 4,4′-(perfluoropropane-2,2-diyl)dibenzoyl chloride (4.2914 g, 10 mmol) solubilized in 5.0 mL of anhydrous DMAc. The reaction mixture was allowed to reach room temperature and kept under constant stirring and N_2_ atmosphere for 24 h. After the reaction time was complete, the reaction mixture was poured into 600 mL of a 10%*w*/*v* NaHCO_3_ aqueous solution, and the resulting precipitate was stirred for 2 h. The polymer was filtered, washed with abundant distilled water, and purified by consecutive solubilization, or precipitation cycles, using DMF and ethanol as the solvent and precipitant, respectively. Finally, the polymer was cleaned in a Soxhlet apparatus using acetone and dried in a vacuum oven at 150 °C until constant weight. The results of the spectroscopic characterization of the polymers (Appendix A) are found in Appendix A.

### 2.4. Preparation of Co-Polyamide Films

The films were obtained through the solvent-casting method. Briefly, 800 mg of the respective co-polyamide was dissolved in 20 mL of DMF (4%*w*/*v*) and filtered through a glass fiber with 3.1 μm porosity. The filtered solutions were deposited on Petri dishes (15 cm in diameter) and kept at 30 °C for 12 h. Subsequently, the obtained films were peeled off from the dishes and placed between two stainless-steel meshes and dried at 150 °C for 24 h [36]. The resulting films exhibited remarkable flexibility and thickness values, ranging between 37.0 and 98.6 μm (see Figure 1).

### 2.5. Preparation of Nonwoven Polyester/Co-Polyamide Membranes

The composite membranes (M-Poly-Ph, M-Poly-Py, and M-Poly-PyM) were manufactured using a 19 cm × 27 cm nonwoven polyester membrane (M) as the support material. In the first step, the nonwoven polyester membrane was placed on a paint applicator (RK K paint applicator) and fixed to the base, as shown in Figure 2.

In parallel, at room temperature, a viscous solution of 2 g of co-polyamide in 10 mL of DMF was prepared and slowly dropped onto the initial part of the applicator while advancing at a speed of 1 cm s^−1^ (step 2). Finally, the composite membrane was allowed to stand for 10 min, then it was picked up and slowly placed into a container with distilled water. The membrane was allowed to stand in the water for 2 h; then, the excess water was allowed to drain off, and it was placed in the drying oven at 40 °C for 24 h. The resulting composite membrane was then sized for the analyses and tests (Figure 3).

### 2.6. Characterization Techniques

#### 2.6.1. Spectroscopic Characterization

The molecular structure of the intermediates, monomers, and co-polyamides was confirmed by ^1^H, ^13^C, ^19^F, DEPT 135°, COSY, HMQC, and HMBC NMR spectroscopy using a Bruker Avance 200 MHz and Bruker Avance III HD-400 spectrometer. Deuterated dimethylsulfoxide (DMSO-*d_6_*) was employed as the solvent and tetramethylsilane (TMS) as an internal reference. FT-IR spectroscopy was performed on a Perkin Elmer Spectrum Two spectrophotometer with a UATR module (ZnSe) in the range of 4000 to 400 cm^−1^ with a resolution of 0.5 cm^−1^.

#### 2.6.2. Molecular Weights

The weight and number average molecular weights of co-polyamides (Mw and Mn, respectively), and the polydispersity index (PDI) were determined using a GPC System 150cv chromatograph (Waters, Milford, MA, USA) equipped with a refractive index detector and a GPC KF-803 column (8 × 300 mm^2^), with a 0.02 M solution of LiBr in DMF as the mobile phase. The system was calibrated with polyethylene glycol oxide standards. The soluble samples (c = 1.0 mg mL^−1^) were filtered through 2 μm microfilters, and then 100 μL was injected at a rate of 1 mL min^−1^.

#### 2.6.3. Mass Density

The density of the co-polyamides was determined from three rectangular sections (2 × 2 cm^2^) of the films fabricated for each sample (Figure 4). The height (*h*) and width (*w*) of the films were measured using a Vernier caliper (0–150 mm, 0.02 mm, UBERMANN), the thickness (*t*) was measured using an electronic outside micrometer (0–25 mm, 0.001 mm, Schut Geometrical Metrology), the mass ($m_{p}$) was measured on an analytical balance (10 mg–220 g, 0.1 mg, AS 220.R2 RADWAG), and the density ($\rho_{p}$) was calculated as indicated in Equation (1) [37].


(1)
$$\frac{m_{p}}{h\times w\times t}=\rho_{p}$$


#### 2.6.4. Mechanical Properties

A tensile test of the co-polyamides (Figure 5) was performed under uniaxial tension on a Shimadzu EZ-LX universal testing machine with a 1000 N load cell at a crosshead speed of 1 mm min^−1^. For this purpose, sample films were cut in rectangular strips (10 × 2.5 cm^2^) with a thickness between 37 and 99 μm, which was measured with an electronic outside micrometer (0–25 mm, 0.001 mm, Schut Geometrical Metrology) [36].

#### 2.6.5. Thermal Properties

The melting point of the intermediates and monomers was determined using a melting point apparatus (Electrothermal, Staffordshire, United Kingdom). For co-polyamides, the thermal stability was determined using a TGA-50 SHIMADZU thermogravimetric analyzer (Columbia, Portland, OR, USA). For this analysis, approximately 2–2.5 mg of the sample was exposed to a temperature between 20 °C and 800 °C with a heating rate of 10 °C min^−1^, under a nitrogen atmosphere. The glass transition temperature (Tg) was established using a PerkinElmer DSC 4000 differential scanning calorimeter from the second scan (PerkinElmer, Hopkinton, MA, USA). The measurements were taken from 15 °C to 415 °C at a speed of 20 °C min^−1^, under a nitrogen atmosphere, with a flow of 20 mL min^−1^.

#### 2.6.6. Surface Properties of Membranes

Scanning electron microscopy (SEM) was performed on a Phenom Prox scanning electron microscope (Thermo Fisher Scientific, Waltham, MA, USA) equipped with a backscattered electron detector and using an accelerating voltage of 15 kV. To achieve this, 10 mm diameter samples of composite membranes and pristine nonwoven polyester membranes were cut.

The contact angles of water (polar solvent) and diiodomethane (non-polar solvent) on the nonwoven polyester membrane and the three composite membranes were measured with a drop shape analyzer (Krüss DSA 255, Hamburg, Germany). For this analysis, 10 drops of 2.0 μL of each solvent were deposited on a 6 × 1.5 cm rectangular section of each sample at a rate of 2.67 μL s^−1^ and 20 °C. Subsequently, the images obtained were processed with the Krüss ADVANCE program using a sessile drop orientation, and the contact angles were determined using the Young–Laplace method. Figure 6 shows, as an example, photographs of a nonwoven polyester membrane (M) and a Poly-Ph-based composite membrane in contact with water and diiodomethane.

From the average contact angles obtained for each liquid–membrane pair, and using the Owens–Wendt–Rabel–Kaelble analysis method (Appendix A of the Appendix A and Equations (2) and (3)), the surface energy ($\gamma_{S}$) and the polar $\gamma_{S}^{p}$ and apolar $\gamma_{S}^{d}$ components (dispersion) of each fabricated membrane were determined [38,39,40,41].

(2)$$\gamma_{S}=\gamma_{\mathrm{SG}}^{p}+\gamma_{\mathrm{SG}}^{d}$$(3)$$(\frac{\gamma_{\mathrm{LG}}(\cos\theta +1)}{2\sqrt{\gamma_{\mathrm{LG}}^{d}}})=\sqrt{\gamma_{\mathrm{SG}}^{d}}+(\frac{\sqrt{\gamma_{\mathrm{LG}}^{p}}}{\sqrt{\gamma_{\mathrm{LG}}^{d}}})\sqrt{\gamma_{\mathrm{SG}}^{p}}$$ where $\gamma_{\mathrm{SG}}^{d}$ and $\gamma_{\mathrm{SG}}^{p}$ are the nonpolar and polar component of the surface energy of a solid, respectively, while $\gamma_{\mathrm{LG}}^{d}$ and $\gamma_{\mathrm{LG}}^{p}$ are the polar and nonpolar components of the surface energy of a liquid.

The membrane porosity was determined by gravimetry on rectangular samples of 1 × 4 cm^2^. For this purpose, the membranes were immersed in distilled water for 24 h. Afterward, the surface of the membrane was dried with absorbent paper, and the wet membrane was weighed (W_w_). The samples were dried at 40 °C under vacuum overnight. After this period, the dry membranes were weighed (W_d_) and the porosity was calculated using Equation (4) [42,43].

(4)$$\mathrm{Porosity}[\%]=\frac{W_{w}-W_{d}}{dA\delta }\times 100$$ where *d* is the water density (1.0 g cm^−3^), *A* is the surface area of the wet membrane (cm^2^), and δ is the thickness of the wet membrane (cm).

#### 2.6.7. Water Flux and Salt Rejection

The water flux and salt rejection of the membranes were determined in a cross-tangential flow filtration system, as shown in Figure 2. To carry out the experiment, the membranes were first compacted with distilled water for 30 min at a pressure of 4 bar. Then, a solution of 30–35 g L^−1^ of NaCl (saline water) was circulated for 60 min at a pressure of 4 bar. At the end of the filtration process, the volume of collected water (filtered water) was measured, and its NaCl concentration was determined.

The NaCl concentration of the saline water and filtered water was determined from a curve obtained between the electrical conductivity and the concentration of four solutions prepared with 10, 20, 30, and 40 g L^−1^ of NaCl.

The water flux (*J*) and salt rejection value (*R_i_*) were performed using Equation (5) and (6), respectively [44].

(5)$$J=\frac{V}{At}$$(6)$$R_{i}=\frac{(C_{s}-C_{f})\times 100}{C_{s}}$$ where *V* = filtered water volume (L), *A* = membrane area (m^2^), t = filtration time (h), *C_f_* is the filtering concentration (g L^−1^), and *C_s_* is the saline water concentration (g L^−1^).

## 3. Results and Discussion

### 3.1. Synthesis and Spectroscopy Characterization of Monomers

The synthesis of the diamine monomers was carried out in two stages. In the first stage, the amidation of 3,5-dinitrobenzoyl chloride with the respective amine (aniline, 4-methylpyridine, and 4-aminomethylpyridine) was performed, using DMF or THF as solvents depending on the solubility of the amine. TEA was used as a base to capture HCl and thus prevent hydrolysis of the bond formed. For the isolation of PhDN, the reaction mixture was concentrated until a viscous mixture was obtained, which was then slowly dropped into distilled water, resulting in a yellow precipitate. This dinitro compound was recrystallized from ethanol. In the case of PyDN and PyMDN, the reaction mixture was concentrated, and the resulting viscous paste was poured over an aqueous NaOH solution to promote precipitation of the products. Finally, the solids obtained were purified by recrystallization using a DMF–water mixture. The yield obtained for these dinitro-amide derivatives ranged from 80 to 96%.

In the second stage, the reduction of the nitro groups was carried out at 60 °C using Pd/C as a catalyst, ethanol as a solvent or dispersant, as applicable, and 80% hydrazine as a hydrogen source. During the addition of hydrazine, a purple solution was observed, which quickly began to turn greenish. After 3 h of a reaction, the mixture was heated at 90 °C to promote the reduction of the remaining nitro groups. After isolation, the aromatic diamine–amide compounds were purified by sublimation, achieving a total yield of 80–85%.

The structural characterization of the precursors and monomers was carried out using infrared (FT-IR-ATR) and nuclear magnetic resonance (NMR) techniques. The results of these analyses are detailed in the Materials and Experimental Methodology section, and the respective spectra are shown in the Appendix A (Appendix A).

For dinitro-amide derivatives, the presence of bands between 3322 and 3096 cm^−1^, 1676 and 1649 cm^−1^, 1534 and 1530 cm^−1^, and 1346 and 1321 cm^−1^ were attributed to the N-H, C=O, N=O asymmetric, and N=O symmetric stretching vibrations, respectively. These observations, along with the signals in the ^1^H NMR spectra assigned to the amide hydrogen (H-6) (9.82–11.15 ppm) and the signals in the ^13^C NMR spectra assigned to the amide carbonyl carbon atom (C-5) (161.12–62.48 ppm), confirm the identity of each compound. The structure of the diamine–amide monomers was initially verified by the presence of the bands observed in the FT-IR spectra, with the most relevant bands corresponding to the symmetric and asymmetric N-H stretching of the amino groups (3312–3412 cm^−1^). Additionally, no bands associated with the nitro group were observed. In the ^1^H NMR spectra, a singlet integrating for 4H between 4.89 and 5.00 ppm was assigned to the amino hydrogen atoms. The ^13^C NMR spectra show a significant alteration in the C-1 chemical shift when comparing a given precursor with its respective diamine derivative, with values ranging from 124.0 to 124.52 ppm for the dinitro compounds and 102.3 to 102.71 ppm for the diamines. All these spectroscopic observations confirm the successful reduction of the nitro-to-amino groups and, therefore, the synthesis of the three target diamines.

### 3.2. Synthesis and Structural Characterization of Co-Polyamides

The co-polyamidations reactions were carried out using the three synthesized diamine (PhDA, PyDA, and PyMDA). Each of these diamines was mixed in an equimolar proportions with ODA and reacted with 4,4′-(perfluoropropane-2,2-diyl)dibenzoyl chloride in a 1:1 molar ratio of diamine to chloride (Figure 3).

At the start of the polymerization, the respective diamine mixtures were completely dissolved in anhydrous DMAc. LiCl was then added, followed by the addition of the acid chloride. All this was carried out at −10 °C to prevent the decomposition of the acid chloride and to maximize the conversion of the respective co-polyamides. After purification through re-precipitation from a DMF-ethanol mixture, followed by Soxhlet extraction with acetone, the co-polyamides were isolated as yellow filaments. The solvents used in these processes were selected based on the solubility tests performed on the unpurified polymers.

The FT-IR-ATR spectra of the three co-polyamides showed bands centered at 3281–3291 cm^−1^, 1650–1653 cm^−1^, and 1205–1208 cm^−1^/1170 cm⁻^−1^, which were attributed to N-H, C=O, and C-F stretching vibrations, respectively (Figure 7). Other band assignments can be found in the Materials and Experimental Methodology section, with the individual spectra for each sample provided in the Appendix A (Appendix A).

NMR analyses were performed in DMSO-*d_6_* because, during the solubility tests, the purified polymers dissolved rapidly in their non-deuterated counterpart, whereas in DMF and THF, the process was slower. The ^1^H NMR spectra revealed two signals for the non-equivalent backbone amide hydrogen atoms. The chemical shift of H-7 appeared between 9.81 and 10.75 ppm, while H-16 was observed between 8.10 and 10.48 ppm. The hydrogen of the amide group in the diamine (H-6) was detected between 9.16 and 10.41 ppm. This distinction was made possible by the difference in signal integration. Due to the stoichiometric ratio used between the diamines and the acid chloride (1:1:2), the skeleton amide signals generally integrate for 4H, whereas the amide group from the diamine moiety integrates for 1H. This was consistent with the integration of the other hydrogen atoms in the polymers, which followed the same ratio as the stoichiometry. The presence of these amide groups was corroborated by ^13^C NMR analysis. The spectra showed three signals between 164.65 ppm and 166.83 ppm. However, unlike the hydrogen assignments, the carbon atoms were not directly assigned from the ^13^C NMR spectra. Instead, the results of the HMBC and HSQC analyses obtained for each polymer were used to assist in the assignments. These results, along with the ^1^H, ^13^C, Dept 135°, COSY, and ^19^F NMR spectra, allowed for the complete assignment of all the nuclei present in the co-polyamides. All spectra are shown in the Appendix A, (Appendix A), and the details of the information obtained are summarized in the Materials and Experimental Methodology section.

### 3.3. Molecular Weights, Density, and Solubility of Co-Polyamides

One of the most important tests performed on the purified co-polyamides was the solubility test, carried out at room temperature. The main results of this test are summarized in Table 1. This test helps determine the ideal solvent for the characterization and processing of the polymers, as well as for the preparation of the composite membranes. The polymers were insoluble in protic polar solvents, as well as in low-polarity or nonpolar solvents. However, like most aromatic polyamides, they were soluble in highly polar solvents such as DMF and DMSO.

In relation to reported aromatic polyamides [29], one of the advancements achieved for this series of co-polyamides was the improvement in solubility in THF, a solvent of intermediate polarity and high volatility. This result could help reduce the costs associated with processing these co-polyamides, particularly in the preparation of coatings via casting or aerosols methods. When comparing the structure of these co-polyamides with that of commercial polyaramids such as Kapton or Kevlar [45], it can be inferred that the increase in the number of segments with a greater degree of freedom—such as carbon and oxygen atoms with sp^3^ hybridization in the main chain—and the inclusion of polar groups like CF_3_ units, contributed to the improved solubility. However, in this case, the increase in free volume contributed by the pendant groups was the most influential factor in the analyzed parameter.

Poly-PyM exhibited the highest dissolution rate in DMF, DMSO, and THF among the series, which can be correlated with its low density (Table 2). Compared to the other two co-polyamides, this polymer displayed the largest interchain space (free volume), which facilitated the penetration of solvents and its subsequent dissolution.

Regarding chain sizes, Poly-PyM has a higher molecular weight than Poly-Ph and Poly-Py (Table 2), indicating that the degree of conversion of the monomers into the respective co-polyamides was higher for this sample. This supports previous observations made in other polymerization processes developed by the group [36,46], where it has been shown that the greater the solubility of the growing polymer in the reaction medium, the higher the degree of conversion and, consequently, the molecular weight. The low polydispersity index (1.03–1.66) calculated for each sample confirms the success of the purification process, with the chains of the polymers having similar sizes on average.

### 3.4. Thermal Properties of Co-Polyamides

Thermogravimetric and calorimetric analyses were performed on the three co-polyamides. Figure 8 shows the thermogravimetric curves and their derivate concerning temperature, while Table 3 summarizes the key results of both analyses.

All three co-polyamides can be considered thermally stable materials, as their degradation temperature, recorded as the 10% mass loss, was above 400 °C, except for Poly-PyM, which showed a T_10%_ value of 382 °C. The high thermal resistance exhibited by this sample is likely related to the high aromatic content in its main chain. Furthermore, in all cases, the onset of thermal degradation occurred near 245 °C. The observed thermal behavior of these co-polyamides indicates a wide temperature window for their use as membranes in water filtration or desalination processes, where the temperatures typically do not exceed 40 °C on average [47].

High residue values after 900 °C were obtained (33–43%). These values are consistent with the high number of heteroatoms in the samples, such as nitrogen, fluorine, and oxygen, which favor the formation of non-volatile residues. In addition, the high values of the temperatures at the maximum degradation rate (T_d1_, T_d2_ and T_d3_) are associated with the presence of the pyridinyl group, and especially the methylpyridinyl group, which promote the cleavage of the bond between the pendant group and the main chain. This may be due to the higher polarity of these groups, and in the case of the methylpyridinyl group, the presence of the aliphatic fragment (CH_2_), which requires less energy for degradation.

The curves obtained from calorimetric analyses performed on the three co-polyamides did not show a glass transition. This could be explained by the high aromatic content in the main chain of the co-polyamide series or by the fact that the glass transition occurs at a temperature close to the degradation point. Regardless, the structural stiffness of the chains, which is influenced by this phenomenon, is favorable for the application of these co-polyamides in the manufacture of desalination membranes. This is because the co-polyamides will not modify their free volume at the temperature of use, thereby increasing their useful life and performance.

### 3.5. Mechanical Properties of Co-Polyamides

Tensile tests were performed on films prepared from each co-polyamide (Table 4). The molecular structure of the samples, particularly the nature of the pendant group, had no significant influence on Young’s modulus, yield strength, tensile strength, or elongation at break. These results indicate that, in general, the co-polyamides exhibited similar mechanical behavior, tending to act as flexible rather than elastic materials. The only noticeable trend related to molecular size was the percentage of elongation. Comparing the molecular weights of each polymer showed that co-polyamides with higher molecular weights tended to exhibit greater elongation. This is attributed to the deformation mechanisms of alignment and chain sliding, both of which favor elongation as the chain size increases [48,49].

### 3.6. Surface Microscopy of Membranes

The SEM inspection of the M-Poly-Ph, M-Poly-Py, M-Poly-PyM, and M membrane surfaces was performed (Figure 9). The images showed that a homogeneous layer of each co-polyamide was successfully deposited on the support matrix. However, some imperfections, likely due to the application process of the co-polyamides on M, were observed (Figure 10 and Appendix A). These imperfections were visible in the images, where the co-polyamide layers on the support matrix could be seen. For instance, in Figure 11B (magnification 8700×), corresponding to the support membrane coated with Poly-Py, the fibers of the support membrane were visible.

In this same image and in others obtained with a magnification of more than 11,500× Appendix A (Appendix A), the presence of small concavities on the surface of the composite membranes was observed. These concavities appeared to resemble pores with a diameter of less than 1 μm. However, upon focusing on the imperfections, it could be concluded that these features are part of the porosity of the membrane, which formed during the water immersion process. This process allows for the solvent used during the coating phase to escape.

To corroborate porosity and pore shape, cross-section images of the composite membranes were taken (Figure 11), showing how the coated polymer formed a reticulated network, leaving interconnected channels. Additionally, it was observed that the solutions permeated the entire support membrane, forming a network of channels through it. From the images and using ImageJ software 1.54k, the average diameter of the pores on the surface was measured. The diameters were 340 ± 0.04 nm for M-Poly-Ph, 517 ± 0.10 nm for M-Poly-Py and 550 ± 0.05 nm for M-Poly-PyM. These results indicate that the membranes have macropores.

### 3.7. Hydrophilic Properties of the Membrane Surface

The surface hydrophilicity of the composite membranes was determined by measuring the contact angle generated when a drop of distilled water and diiodomethane was deposited on the sample (Appendix A, Appendix A). Both analyses were carried out at an average temperature between 20 and 25 °C. The results are shown in Figure 12 and Table 5, which also include the surface energy calculations and the porosity percentage for each group of membranes.

As a result of the contact angles recorded, the deposition of the polymers on the nonwoven polyester support membrane generates a surface with significantly increased hydrophilicity. This is attributed to the large number of hydrogen bonds that can be formed by the amide groups created during polymerization and the pyridinyl groups present in the pendant groups of Poly-Py and Poly-PyM.

Among the three groups of membranes, M-Poly-Py was found to be the most hydrophilic. This hydrophilicity is likely due to the ratio between the number of amide groups and the hydrophobic regions (aromatic and aliphatic segments). In this regard, Poly-Py has the highest number of amide groups relative to the hydrophobic regions. It is important to note that although these membranes can be considered hydrophilic, since they present contact angles less than 90°, their hydrophobic component remains high. This is evident from the values of the dispersion component and the polarity of the surface energy of each membrane, where the dispersion component notably predominates.

The porosity percentage determined for the membranes shows that when co-polyamide coating is applied on the support membrane, the porosity decreases significantly. This decrease in porosity is understood to be due to fact that the pores of the support membrane have been filled by the coated polymer. Furthermore, among the three co-polyamides, Poly-Ph resulted in the greatest reduction in porosity compared to the support membrane, indicating that it best penetrated the support matrix. This result could be explained by the higher affinity between Poly-Ph and the support membrane, as it is the most hydrophobic co-polyamide. This is consistent with the fact that M-Poly-Py has the highest percentage porosity and Poly-Py is the most hydrophilic sample. These results, along with those from the contact angle measurements, are expected to directly influence both the rejection behavior and water flow during the filtration process. Based on these findings, M-Poly-Py is anticipated to show higher flux and rejection during filtration tests, while M-Poly-Ph is expected to be the lowest-performing membrane.

### 3.8. Evaluation of Membranes Performance

Membrane performance was evaluated using NaCl solution (35 g L^−1^) and the tangential filtration system described in the Water Flux and Salt Rejection section. The tests were conducted for 1 h at an inlet pressure of 4 bars. During the process, the filtered water was collected to determine the resulting flow rate and, the electrical conductivity was measured to determine the percentage of rejection achieved. The results are summarized in Table 6. Three samples of the support membrane (M), without co-polyamide, were tested. Brine was passed through each sample for 30 s. Between 680 mL and 690 mL of filtered water was collected in each test, yielding a flow rate of 4896 L h^−1^, equivalent to 226,470 L m^−2^ h^−1^. Moreover, the electrical conductivity of the brine and the filtered water was measured, showing no difference between the two.

These results showed that the support membrane failed to reject NaCl during the filtration process due to its low resistance to water flow, owing to its high porosity. Furthermore, the flow rate achieved was determined solely by the size of the tangential section used, which reduced the flow rate, although this was significantly higher than that of the polyamide-coated membranes. Among the three composite membranes, M-Poly-Ph showed zero water flow rate. Despite having a certain percentage of porosity, its low value and lower hydrophilic character prevented water from passing through after 1 h of operation.

M-Poly-Py demonstrated the best performance among the composite membranes due to its higher porosity and hydrophilicity, especially when compared to M-Poly-PyM. This is likely a result of the amide and pyridinyl groups present in its structure. In comparison, M-Poly-Py showed greater porosity than M-Poly-PyM and M-Poly-Ph, which facilitated the free flow of water molecules. Its higher hydrophilicity also promoted the formation of a water layer on the membrane surface [50], which acted as a selective barrier, allowing water to pass while preventing the passage of chloride and sodium ions. This effect contributed to the enhanced rejection performance of M-Poly-Py.

During the design of the co-polyamides, it was anticipated that M-Poly-PyM would exhibit a better rejection-flux ratio due to its higher porosity, resulting from the larger free volume provided by the pendant group. However, the presence of the methylene group, which reduces hydrophilicity and the lower porosity achieved during membrane fabrication compared to M-Poly-Py led to an unsatisfactory evaluation. Nevertheless, the results were promising, as a homogeneous co-polyamide deposit was formed on the surface, and the hydrophilicity of the support membrane was enhanced. Notably, a flux of 17.7 L m^−2^ h^−1^ or 44.3 L m^−2^ h^−1^MPa^−1^ and a NaCl rejection of 37.3% were obtained. Both values are promising, considering that the flux of commercial membranes typically ranges from 35–112 L m^−2^ h^−1^ at 10 bars of pressure [51], with rejection rates ranging from 20% to 99.7% [52]. It is important to note that the lower rejection rate observed, in comparison to commercial membranes, is due to the difference in porosity. The pore size of the membranes obtained ranged from 340 to 550 nm, while commercial membranes capable of rejecting up to 99% of NaCl had pore sizes between 1 and 10 nm. These membranes are generally classified as reverse-osmosis membranes.

The coating process of the co-polyamides can still be optimized, and the porosity of the resulting composite membranes can be better controlled. Additionally, structural modifications can be made to these co-polyamides, particularly to Poly-Py, to improve their hydrophilicity. In water purification processes such as desalination, a highly hydrophilic membrane is preferred, as it promotes wetting and water flow. However, a membrane that is excessively hydrophilic or superhydrophilic tends to lose mechanical stability due to increased wetting, rendering it unsuitable for use. Therefore, the membrane must have a certain level of hydrophobicity to reduce wetting and maintain mechanical stability. Nonetheless, if the hydrophobicity is too high, it could lead to the adhesion of organic matter or the formation of biofilms on the membrane surface (fouling). Striking the right balance between hydrophilicity and hydrophobicity will be specific to the type of membrane manufactured and will be closely linked to the chemical structure of the membrane and the nature of the water being treated.

In the case of these membranes, an optimal ratio has not yet been achieved, as the hydrophilicity of the membranes can still be enhanced through chemical modification of the pyridinyl groups, potentially reducing hydrophobicity. In addition, the impact of hydrophobicity on membrane fouling has not yet been determined, and this will be investigated in the second part of the research.

## 4. Conclusions

A series of three new aromatic co-polyamides were successfully synthesized, combining hexafluoroisopropyl, oxyether, phenyl and amide groups in the main chain, with phenyl, pyridinyl or methylpyridinyl units as pendant groups. The combination of structural elements allowed for the modulation of the polymers’ properties, resulting in good solubility in aprotic polar solvents such as DMF and DMSO, thermal resistance above 350 °C, the ability to form mechanically resistant films, and moderate hydrophilicity, classifying them as hydrophilic materials. Moreover, these co-polyamides were used to fabricate membranes suitable for filtration or water desalination processes, where the flow of filtered water and NaCl rejection could be modulated depending on the pendant group used, reaching values up to 17.7 L h^−1^m^−2^ and 37.3%, respectively.

The performance of these membranes can be enhanced by incorporating structural modifications to further increase hydrophilicity and/or by controlling various variables during the membrane fabrication process. This would enable the control of properties such as porosity percentage and pore size. These aspects present significant opportunities for the further research, development, and exploration of new applications for this series of co-polyamides and their derivatives.

## Data Availability

Data are contained within the article.

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
