# Peer review of "Novel Co-Polyamides Containing Pendant Phenyl/Pyridinyl Groups with Potential Application in Water Desalination Processes"

_polymers, 2025, doi:10.3390/polym17020208_

Round 1
Reviewer 1 Report
Comments and Suggestions for Authors
The manuscript “Novel Co-Polyamides Containing Pendant Phenyl/Pyridinyl Groups with Potential Application in Water Desalination Processes” demonstrates a well-structured experimental approach, from monomer synthesis to membrane performance evaluation. The text is generally clear but includes minor typographical errors (e.g., "pendat group" in the abstract keywords).
How do the authors justify the relatively low NaCl rejection rates (e.g., 37.3% for M-Poly-Py) compared to commercial membranes that achieve >99% rejection?
Could the authors explain the high residue percentages in thermogravimetric analysis? Do these residues affect membrane recyclability?
The contact angle data indicate a significant increase in hydrophilicity after co-polyamide deposition. How does this improved hydrophilicity correlate with the membranes salt rejection performance?
The study mentions hydrophilic properties but notes that the membranes remain partially hydrophobic. Could the authors discuss the ideal hydrophilicity-hydrophobicity balance for desalination membranes and how their membranes compare?
What is the explanation for the significant differences in flux and rejection between the M-Poly-Py and M-Poly-PyM membranes?
Comments on the Quality of English Language
Typographical errors should be corrected
Reviewer 2 Report
Comments and Suggestions for Authors
This is a very good study about membrane film. The author investegated the effect of pendent group by synthesized 3 different benzamines and copolymerize them with polyamide. I suggest to accpet with minor correction.
1. I prefer to move the NMR result from section 2.2-2.3 to supporting information to save space for the main text while interested reader can check the details. Thoes NMR results look good to me.
2. Line 231 "during 24h" should be "for 24h".
3. Line 263 Suggest to change to "mass density" to better deferenciate with others, like hole density or viod density.
4. Figure 7 Suggest to move to SI since this is a well-known knowledge. And the author did not modify it.
5. For all figures, check the bold title again. Some of the title do not have bold font as the others.
6. Scheme 3, what is the ratio of n and m in the copolymer?
